



# The application of mean averaging kernels to mean trace gas distributions

Thomas von Clarmann and Norbert Glatthor

Karlsruhe Institute of Technology, Institute of Meteorology and Climate Research,
Karlsruhe, Germany

**Correspondence:** thomas.clarmann@kit.edu

**Abstract.** To avoid unnecessary data traffic it is sometimes desirable to apply mean averaging kernels to mean profiles of atmospheric state variables. Unfortunately, application of averaging kernels and averaging are not commutative in cases when averaging kernels and state variables are correlated. That is to say, the application of individual averaging kernels to individual profiles and subsequent averaging will, in general, lead to different results than averaging of the original profiles prior to the

application of the mean averaging kernels unless profiles and averaging kernels are fully independent. The resulting error, however, can be corrected by subtraction of the covariance between the averaging kernel and the vertical profile. Thus it is recommended to calculate the covariance profile along with the mean profile and the mean averaging kernel.

## 1  Introduction

More often than not satellite data retrievals are constrained because the unconstrained profile retrieval on a given altitude grid

would lead to an ill-posed inverse problem. The constrained retrieval is more robust but the price to pay typically is, among other effects, a certain loss in vertical resolution. The effect of the constraint is characterized by the averaging kernel matrix (Rodgers, 2000).

Many applications of remotely sensed data involve comparison with independent model or independent measurement data. If these comparison data are better resolved than the remotely sensed data, the averaging kernel of the latter has to be applied

to the former to make the comparison meaningful (Connor et al., 1994). Otherwise differences caused by the different altitude resolution would mask scientifically significant differences. Unfortunately, for a vertical profile of $n$ values of an atmospheric state variable, the related averaging kernel matrix is of the size $n \times n$, this is, the data traffic is dominated by the averaging kernel data while the data product of interest, *viz.* the profile, could be communicated with much less effort. Often the data users are not interested in the individual measurements but prefer to work, e.g., with monthly zonal mean profiles (e.g. Hegglin

and Tegtmeier, 2011; von Clarmann et al., 2012; Hegglin et al., 2013; Hegglin and Tegtmeier, 2017; Tegtmeier et al., 2013, 2016). In this case, it would be convenient if the data user could simply apply monthly zonal mean averaging kernels to her better resolved monthly zonal mean data to make them comparable to the coarser resolved zonal monthly mean measurements. Unfortunately averaging and application of the averaging kernel are not commutative. As soon as the data and the averaging kernels covary, the application of the mean averaging kernel to mean profiles gives a different result than the application of





individual averaging kernels prior to averaging. We solve this problem by providing statistically inferred covariance terms which can be used to correct the related error. In the next section we describe the theoretical framework used. As a case study, covariances applicable to trace gas profiles retrieved from MIPAS (Michelson Interferometer for Passive Atmospheric Sounding, Fischer et al. 2008) measurements are inferred in Section 3. The varying importance of the covariance effect is

illustrated in Section 4. Section 5 is an interlude where we investigate into pitfalls regarding the applicability of averaging kernels to comparison data, before a critical discussion of the applicability of our suggested appproach concludes the paper (Section 6).

## 2    The formal concept

We borrow the formal concept of retrieval theory from Rodgers (2000). The intended application of our study is, at worst,

moderately nonlinear retrievals. That is to say, linear theory is assumed to be adequate for the characterization of the retrieval in terms of error estimation, assessment of vertical resolution, and so forth. Thus, we ignore all complication which may arise from non-linearity and thus do not discuss the retrievals in an iterative setting.

      The vertical resolution of a profile of an atmospheric state variable, e.g., temperature or the volume mixing ratio of a trace gas, with $n$ gridpoints, is usually characterized by the averaging kernel matrix $\mathbf{A}$ of size $n \times n$. Its elements are the partial

derivatives $\frac{\partial \hat{x}_i}{\partial x_j}$ of the estimated state variables $\hat{x}_i$ with respect to the true state variable $x_j$. While the indices $i$ and $j$ typically run over altitude levels of one vertical profile, the concept as such has a much wider range of applicability, e.g., horizontal averaging kernels (von Clarmann et al., 2009a), or characterization of cross-dependence of multiple species. In this study, we restrict ourselves to averaging kernels of vertical profiles of single species. For a constrained retrieval of the type

$$\hat{x} = x_{\mathrm{a}} + \left(\mathbf{K}^T \mathbf{S}_{\mathrm{y}}^{-1} \mathbf{K} + \mathbf{R}\right)^{-1} \mathbf{K}^T \mathbf{S}_{\mathrm{y}}^{-1} \left(y - F(x_{\mathrm{a}})\right), \tag{1}$$

where the $\hat{x}$ vector represents the estimated profile, $x_{\mathrm{a}}$ is an a priori profile, $\mathbf{K}$ is the Jacobian matrix $\frac{\partial y_i}{\partial x_j}$, $^T$ indicates a transposed matrix, $\mathbf{S}_{\mathrm{y}}$ is the measurement error covariance matrix, $\mathbf{R}$ is a regularization matrix, $F$ is the radiative transfer function, and $y$ is the vector of measurements (von Clarmann et al. 2003a, building largely upon Rodgers 2000), the averaging kernel matrix is

$$\mathbf{A} = \left(\mathbf{K}^T \mathbf{S}_{\mathrm{y}}^{-1} \mathbf{K} + \mathbf{R}\right)^{-1} \mathbf{K}^T \mathbf{S}_{\mathrm{y}}^{-1} \mathbf{K}. \tag{2}$$

With this, Eq. 1 can be rewritten as

$$\hat{x} = (\mathbf{I} - \mathbf{A}) x_{\mathrm{a}} + \mathbf{A} x \tag{3}$$

The most common application of the averaging kernel matrix is the degradation of high-resolved vertical profiles to make them comparable to poorer resolved profiles (Connor et al., 1994).

$$x_{\mathrm{degraded}} = (\mathbf{I} - \mathbf{A}) x_{\mathrm{a}} + \mathbf{A} x_{\mathrm{original}} \tag{4}$$



For applications where the a priori profiles are all zero, as being the case for most trace gas profiles retrieved from MIPAS (von Clarmann et al., 2009b), which often is appropriate if a smoothing regularization (Steck and von Clarmann 2001, building on Tikhonov 1963) is used instead of an inverse a priori covariance matrix as suggested by Rodgers (1976, 2000), this reduces to

$$\boldsymbol{x}_{\mathrm{degraded}} = \mathbf{A}\boldsymbol{x}_{\mathrm{original}}. \tag{5}$$

Calculation of zonal averages over $L$ profiles renders[1]

$$
\begin{aligned}
<\hat{\boldsymbol{x}}> \quad &= \quad <(\mathbf{I}-\mathbf{A})\boldsymbol{x}_{\mathrm{a}}+\mathbf{A}\boldsymbol{x}> \\
&= \quad <\boldsymbol{x}_{\mathrm{a}}> - <\mathbf{A}><\boldsymbol{x}_a> - \\
&\quad\quad \mathrm{cov}(\mathbf{A},\boldsymbol{x}_{\mathrm{a}}) + <\mathbf{A}><\boldsymbol{x}> + \mathrm{cov}(\mathbf{A},\boldsymbol{x}),
\end{aligned}
\tag{6}
$$

where

$$\mathrm{cov}(\mathbf{A},\boldsymbol{x}) = \frac{1}{L}\sum_{l=1}^{L}(\mathbf{A}_l - <\mathbf{A}>)(\boldsymbol{x}_l - <\boldsymbol{x}>), \tag{7}$$

and *mutatis mutandis* for $\mathrm{cov}(\mathbf{A},\boldsymbol{x}_{\mathrm{a}})$.

For a retrieval with $\boldsymbol{x}_{\mathrm{a}} = \mathbf{0}$ this simplifies to

$$
\begin{aligned}
<\hat{\boldsymbol{x}}> \quad &= \quad <\mathbf{A}\boldsymbol{x}> \\
&= \quad <\mathbf{A}><\boldsymbol{x}> + \mathrm{cov}(\mathbf{A},\boldsymbol{x}).
\end{aligned}
\tag{8}
$$

$\mathrm{cov}(\mathbf{A},\boldsymbol{x})$ can be approximated by $\mathrm{cov}(\mathbf{A},\hat{\boldsymbol{x}})$ which can easily be evaluated statistically from the available results and distributed to the data user along with the mean averaging kernel $<\mathbf{A}>$ and the mean profile $<\boldsymbol{x}>$ and used to correct profiles of averaged comparison data. All this is valid only *cum grano salis*. Related problems will be discussed in Section 6.

For a retrieval with constant climatological $\boldsymbol{x}_{\mathrm{a}}$ for the entire sample of profiles we get

$$
\begin{aligned}
<\hat{\boldsymbol{x}}> \quad &= \quad <(\mathbf{I}-\mathbf{A})\boldsymbol{x}_{\mathrm{a}}+\mathbf{A}\boldsymbol{x}> \\
&= \quad \boldsymbol{x}_{\mathrm{a}} - <\mathbf{A}>\boldsymbol{x}_a + \\
&\quad\quad + <\mathbf{A}><\boldsymbol{x}> + \mathrm{cov}(\mathbf{A},\boldsymbol{x}).
\end{aligned}
\tag{9}
$$


For a retrieval where an individual prior $\boldsymbol{x}_a$ is used for each profile retrieval, it may also be adequate to assume

$$\mathrm{cov}(\mathbf{A},\boldsymbol{x}) \approx \mathrm{cov}(\mathbf{A},\boldsymbol{x}_{\mathrm{a}}) \tag{10}$$

---

[1]Here a caveat is in order. The average of profiles which are 'optimal' in the sense of maximum a posteriori information and where the a priori information is the same for all averaged profiles is not the optimal average. This is, because the weight of the a priori information will be too large in the average. A more thorough discussion of this issue, however, is beyond the scope of this paper.





and the correction by the covariance terms becomes approximately obsolete, because

$$< (\mathbf{I} - \mathbf{A})\boldsymbol{x}_\mathrm{a} + \mathbf{A}\boldsymbol{x} > \quad = \tag{11}$$

$$< \boldsymbol{x}_\mathrm{a} > - < \mathbf{A} >< \boldsymbol{x}_a > - \mathrm{cov}(\mathbf{A}, \boldsymbol{x}_a)$$

$$+ < \mathbf{A} >< \boldsymbol{x} > + \mathrm{cov}(\mathbf{A}, \boldsymbol{x}) \quad \approx$$

$$< \boldsymbol{x}_\mathrm{a} > - < \mathbf{A} >< \boldsymbol{x}_a > - \mathrm{cov}(\mathbf{A}, \boldsymbol{x}_a) +$$

$$< \mathbf{A} >< \boldsymbol{x} > + \mathrm{cov}(\mathbf{A}, \boldsymbol{x}_\mathrm{a}) \quad =$$

$$< \boldsymbol{x}_\mathrm{a} > - < \mathbf{A} >< \boldsymbol{x}_a > + < \mathbf{A} >< \boldsymbol{x} >$$

For retrievals performed in the log-space, all this becomes slightly more complicated (e.g., Stiller et al., 2012). Equation (4) then reads

$$\boldsymbol{x}_\mathrm{degraded} = \exp\left((\mathbf{I} - \mathbf{A})\ln\boldsymbol{x}_\mathrm{a} + \mathbf{A}\boldsymbol{x}_\mathrm{original}\right), \tag{12}$$

where $\mathbf{A}$ is $\frac{\ln \hat{\boldsymbol{x}}_i}{\ln \boldsymbol{x}_j}$. For log retrievals there is no obvious way to correct for the averaging artefacts as long as the averaging is performed linearly in the vmr-space. Since averaging of logarithmic rerievals in the logarithmic domain has its own problems (Funke and von Clarmann, 2012), we do not pursue this option any further.

The issues discussed in this Section have to be considered if mean averaging kernels are to be applied to mean profiles in the spirit of Eq. 4, in order to make mean profiles of different sources comparable.

## 3 Covariances

The covariances between the averaging kernel matrices and the state vectors are calculated as

$$\mathrm{cov}(\mathbf{A}, \boldsymbol{x}) = \tag{13}$$

$$= \frac{1}{L}\left(\sum_{l=1}^{L}(\mathbf{A}_l - < \mathbf{A} >)(\boldsymbol{x}_l - < \boldsymbol{x} >)\right) =$$

$$= \frac{1}{L}\left(\sum_{l=1}^{L}\mathbf{A}_l\boldsymbol{x}_l - \frac{1}{L}\sum_{l=1}^{L}\mathbf{A}_l\sum_{l=1}^{L}\boldsymbol{x}_l\right),$$

where $L$ denotes the sample size; we divide by $L$ instead of $L-1$ because the latter would entail an inconsistency with Eq. (6) and Eqs. (8–11). The formulation in the lowermost line of Eq. (13) is computationally more efficient. For our case study, averaging kernel matrices and state vectors retrieved from limb emission spectra measured by the MIPAS are used. The general processing scheme is described by von Clarmann et al. (2003b, 2009b). We study covariances for MIPAS $O_3$ and HCN profiles (Laeng et al. 2018 and Glatthor et al. 2015, respectively).

To illustrate the relevance $\tilde{r}$ of the correction terms, we also present the normalized covariance term for each profile element:

$$\tilde{r}_n = \mathrm{cov}(\mathbf{A}, \boldsymbol{x})_\mathrm{n}/(< \mathbf{A} >< \boldsymbol{x} >)_\mathrm{n} \tag{14}$$





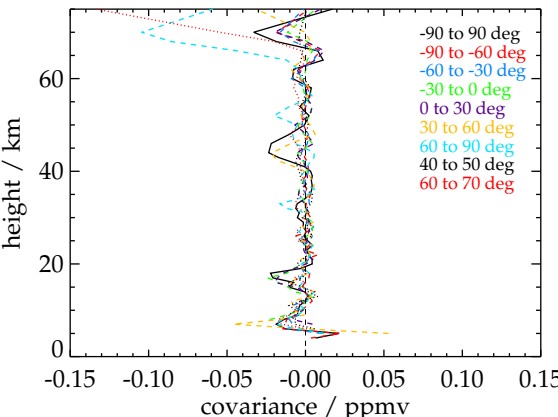

**Figure 1.** : Covariance of the averaging kernel and ozone mixing ratio for various latitude bands. The black solid line refers to global data. The dashed lines refer to 30-degrees latitude bands, and the dotted lines to 10-degrees latitude bands.

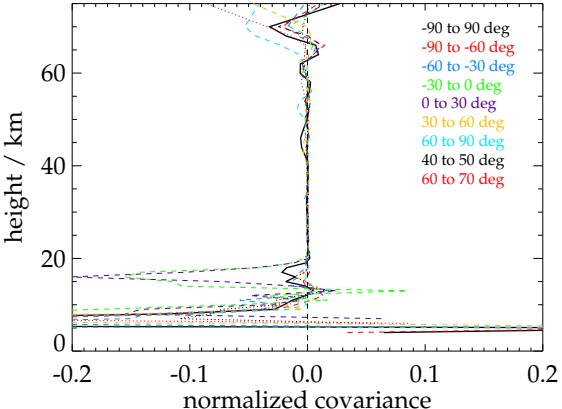

**Figure 2.** : Correlation of the averaging kernel and ozone mixing ratio for various latitude bands.

where index $n$ runs over the profile elements. The ˜ symbol shall avoid confusion with the product moment correlation co-efficient established by Pearson (1895) for which $r$ is often used as a symbol and which is widely used for normalization of covariances but which causes some headache when applied to correlations of matrices with vectors. For simplicity, we still call the normalized covariance 'correlation', however without claiming equivalence with its scalar counterpart.

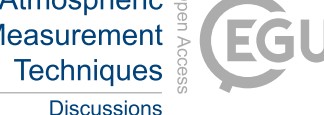



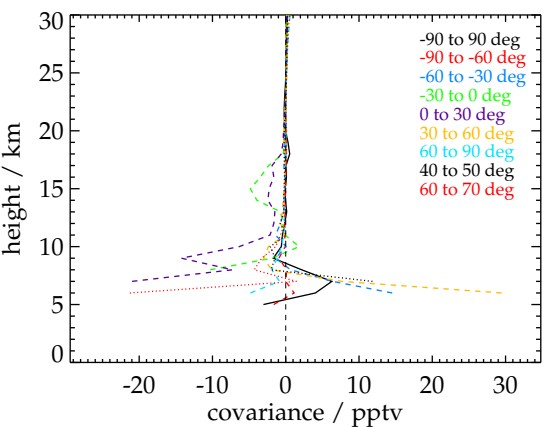

**Figure 3.** : As Fig 1, but for HCN.

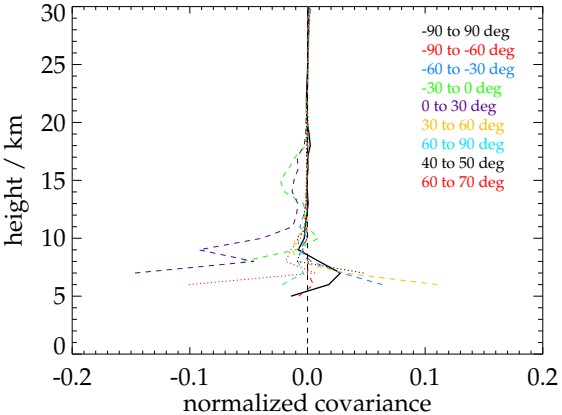

**Figure 4.** : As Fig 2, but for HCN.



## 4  Results

Case studies have been performed using ozone and HCN vertical profiles retrieved from MIPAS measurements of February 10, 2009. The test data set consistes of 1385 geolocations. This day was characterized by a significantly disturbed Arctic vortex. Figure 1 shows the covariances between the profiles and the averaging kernel matrices of ozone globally (black solid line) and for various latitude bands of different size (dashed and dotted lines). In general the values are largest at the extreme ends of the profiles, where the effect of the constraint on the retrieved profile is typically largest.

These results suggest that for MIPAS ozone in the middle and upper stratosphere the effect studied here can be safely ignored. Problems are limited to the upper troposphere and lower stratosphere and the mesosphere. The relevance of this effect can better be judged on the basis of the correlation profiles (Fig. 2). From 20 to about 60 km the effect is negligibly small for all latitude bands investigated in this case study. Only at the uppermost and lowermost altitudes the effect becomes relevant. The large effects at lower altitudes are simply caused by normalization of the original covariances by low ozone mixing ratios.

To study HCN is particularly interesting in the tropical upper troposphere and lower stratosphere. This is because HCN has tropospheric sources, and its pathway into the stratosphere is a particular research issue. The covariance effects can exceed 10% (violet and yellow dashed lines) and thus need to be considered when mean profiles are used for quatitative analysis and mean averaging kernels are applied.

These case studies are not meant to be representative for other gases or other instruments. Instead, they are shown to give a flavour about the order of magnitude this kind of effect can reach. Unless $\mathrm{cov}(\mathbf{A}, \boldsymbol{x})$ can be shown to be small, we recommed to use this covariance term for an additive correction when mean averaging kernels are applied to averaged comparison data.

## 5  An Important Side Remark

The issue of the limited applicability of averaging kernels to independent comparison data deserves awareness. When averaging kernels of a measurement are applied to better resolved comparison data, it is almost always tacitly assumed that the atmospheric state represented by the measurement is the same as that of the comparison data and that thus the averaging kernel of the measurement can be safely applied to the comparison data. However, since averaging kernels are in general state-dependent, a caveat is in order.

Application of the formalism of Connor et al. (1994) (our Eq. 4) has its own specific problems which fully apply to our proposed scheme. The application of the averaging kernel matrix of a poorly resolved profile $\boldsymbol{x}_{\mathrm{coarse}}$ to a better resolved profile $\boldsymbol{x}_{\mathrm{fine}}$ is only adequate if both data sets describe approximately the same atmospheric state, i.e., if the one profile is in the linear domain of the other. That is to say, if the same Jacobians apply to both profiles. Otherwise it would be necessary to construct an averaging kernel using the Jacobian $\mathbf{K}$ evaluated for the atmospheric state represented by the profile $\boldsymbol{x}_{\mathrm{fine}}$ but with the measurement covariance matrix $\mathbf{S}_{\mathrm{y}}$ and the regularization matrix $\mathbf{R}$ corresponding the the retrieval producing $\boldsymbol{x}_{\mathrm{coarse}}$. Within linear theory, the state-dependence of the Jacobian $\mathbf{K}$ and with this the state dependence of the averaging kernel matrix $\mathbf{A}$ are often ignored. To do so is justifiable as long as the profiles to be intercompared are sufficiently similar. In this case the comparison will show reasonable agreement.





If, in turn, the profiles are very different, two components contribute to the disagreement seen after application of the Connor-method. First the genuine difference of the profiles, and second any artefact caused by the inadequate averaging kernels. Thus, in the logic of a testing scheme, good apparent agreement hints at good genuine agreement *a fortiori*, because it is extremely unlikely that genuine differences which would survive the application of the Connor-method with the correct averaging kernel

are 'convolved away'[2] with the averaging kernel evaluated for the wrong atmosphere.

## 6    Discussion and Conclusion

We have identified the problem that it is not generally allowable to apply mean averaging kernels to mean atmospheric profiles in situations where the averaging kernels and the profiles covary. The relevance of this effect, however, depends on the instrument, the species, the latitude band and the altitude under investigation. To solve this problem, we have proposed a statistical

correction scheme which involves the covariance between the averaging kernel and the profile. With this correction in place, the scheme suggested by Connor et al. (1994) to make better resolved vertical profiles of atmospheric state variables comparable to coarser resolved ones can be applied also to averaged profiles.

For data producers who distribute, besides their original retrievals, also zonal mean data or similar data products, we recommend the following: Along with the generation of zonal mean data and averaging kernels, the correlation profiles should

be calculated. Compared to averaging kernels and covariance matrices they need negligible storage and cause negligible data traffic. In cases when zonal mean data have already been generated but when mean covariance matrices and covariance profiles are not available, the huge I/O load associated with reading all individual averaging kernels may be prohibitive. In these cases one might consider to estimate the mean averaging kernel and the covariance profile on the basis of a limited random sample out of the measurements which went into the zonal mean.

*Data availability.*   MIPAS data used for this case study are available via https://www.imk-asf.kit.edu/english/308.php

*Author contributions.*   TvC identified the problem, suggested the research method, proposed the solution and wrote the paper. NG critically reviewed the methodology, implemented the suggested solution as a computer program, performed the calculations and visualized the results. Both authors discussed and evaluated the results and elaborated the recommendation.

*Competing interests.*   TvC is associate editor of AMT but has not been involved in the evaluation of this paper.

---

[2]We put this term in quotes to highlight that this method is, mathematically speaking, not a convolution. It is not even a numerical approximation to a convolution.



*Acknowledgements.* NG acknowledges support by BMBF project SEREMISA (50EE1547). SPARC provided funding for the meetings of the activity "Towards Unified Error Reporting (TUNER)", the framework in which this research took place. The International Space Science Institute (ISSI) provided support and facilities for two TUNER International Team meetings.



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
