# Peer review of "The application of mean averaging kernels to mean trace gas distributions"

_Atmospheric Measurement Techniques, 2019_

## Referee Comment (RC1) · Anonymous Referee #1 · 7 Apr 2019

Referee comments on

The Application of mean averaging kernels to mean trace gas distributions. Clarmann & Glatthor Submitted to Atmos. Meas. Tech., 4 March 2019

Overview

The paper draws attention to a hitherto overlooked problem with the application of averaging kernel matrices, specifically that the AK matrix itself, A, has some dependence on the retrieved state, x. Hence, when using averaged data, <x> (e.g., a monthly mean), the appropriate averaging kernel <Ax> is not simply <A><x> constructed from an average AK matrix <A>.

Main Comments

[Figure]

The paper is concise, well-argued and includes a suitable illustrative example, although I have some suggestions below as to how it might be further clarified so as to reach a wider audience.

While the main recommendation of the paper is that the data providers should also provide a correction term for the mean averaging kernel, it does seem more practical if, instead, the data providers themselves use the guidance in this paper to produce a suitable averaging kernel to accompany the averaged products. We all know that it is a struggle to get data user to understand and apply an averaging kernel, so we should avoid making their task any more complicated.

I doubt if this paper will be the last word in the matter - there are a number of open issues which require a little more thought, such as logarithmic retrievals, retrievals of temperature/pointing/pressure, non-constant a priori data, averaging kernel matrices which are not square. However, this paper is a good starting point for the conversation.

Minor Comments

1) The authors frequently resort to Latin. Personally I find it a welcome change from the usual stock phrases, although I expect some readers may not be quite so appreciative.

2) Abstract (and elsewhere): reference to 'covariance profile' although the suggested correction is a matrix rather than a profile.

3) P1, L18: 'this is' does not make sense here.

4) P1, L20: I don't think the use of monthly means requires any references, although no doubt Hegglin and Tegtmeier will appreciate being selected for multiple citations from among many, many such users.

5) P1, L20: suggest 'their' rather than 'her'.

6) P2, L13: The casual reader may interpret this comment as suggesting none of this applies to non-linear, iterative retrievals, so I suggest rewording to emphasise that it

still does.

7) P2, Eq 2: It could be pointed out that the main dependence on x in the AK matrix comes from the Jacobian matrix, K (although possibly also from R if some form of adaptive regularization is used), so whether or not there is any dependence of A on x is usually a consequence of whether or not K depends on x.

8) P2, L25: An extra equation, y-F(x_a) = K ( x - x_a ) would help the reader get from eq (2) to eq (3).

9) P2, L28 onwards. This is confusing. Elsewhere averaging kernels are discussed as a characteristic of the lower-resolution (satellite) retrievals, but in this example (Eq 4) the averaging kernel seems to be on the grid of the higher resolution 'original' retrieval. Despite the similarity of Eq 4 and Eq 3, these seem to be two quite different things.

10) P3, Eq 6 and elsewhere: if this is prepared with LaTeX, I suggest using \langle and \rangle rather than < and > for the angle-brackets.

11) P3, Eq 7: it would be helpful to further simplify this here, giving cov(A,x) = < A x > - <A> <x> which makes Eq 6 clearer.

12) It does not help that many of these equations are split over two lines, but that may not be the choice of the authors.

13) P3, L22: There seems to be more to be said than this simple phrase 'an individual prior x_a'. For example, an individual but *almost* constant a priori could be used for each profile, in which case Eq (9) applies rather than Eq (11). The key is obviously what sort of 'individual prior' leads to the two covariances being approximately equal.

14) P4, Eq (11): In this case I think the extra equation confuses (especially when split over multiple lines) rather than clarifies. Perhaps better to refer back to Eq (6) and simply state the simplified result.

15) P4, L26: \bar r seems to be introduced in the wrong place in this sentence, presumably it should be after 'normalized covariance term'

16) P4, L14: I'm surprised that this produces stable results, eg for HCN at higher altitudes, where the <x> in the denominator would tend to zero. Covariance terms, as in the Pearson correlation, are usually scaled by the square root of the variance, so don't have this problem.

17) P5, L17: 'recommend' (spelling).

---

## Referee Comment (RC2) · Anonymous Referee #2 · 10 Apr 2019

This manuscript discusses an important and often ignored issue involving the application of averaging kernels to mean profiles. A solution to the problem is presented where the covariance between the averaging kernel and the atmospheric state is calculated. Examples are shown applying the method to MIPAS, and recommendations are given to data producers of monthly zonal mean data.

The manuscript is well written and suitable for publication in AMT after a few comments are taken into account.

[Figure]

**General Comments**

The discussion and conclusion (including the recommendations) of the paper focuses on the ideal case where the data producer actually calculates (and stores) an averaging kernel for each individual profile. It is somewhat common to only produce representative averaging kernels and perhaps use them as a metric for retrieval performance in a validation/retrieval paper or data quality document. Would a possible recommendation of this work be that a few of these covariance terms should be calculated and included as an assessment of the data quality?

Related to the above point, I have to wonder, is the covariance profile useful beyond a correction when applying the mean averaging kernel? My (perhaps wrong) interpretation is that when the covariance profile is 0, the mean of the retrieved profile is a smoothed version of the true mean atmospheric state. I suppose what I am asking is that if the covariance profile is not 0, is it wrong to interpret the retrieved mean as a smoothed version of the true atmospheric mean? If so, I would like to see a discussion of this included in the manuscript.

**Minor Comments**

p.1 l.9: ". . . on a given altitude grid . . . "
Here and throughout this section it is written that altitude is the vertical coordinate, however all of the arguments should equally apply to any vertical coordinate.

p.2 l.18: "For a constrained retrieval of the type"
The way this is presented the reader may assume that what follows only applies to

retrievals applying a (possibly iterative) form of eq. 1, when the concepts here are more general.

p.2 l.29: eq. 4
Somewhere here I would like to see a brief mention that $x_{original}$ needs to be converted to the same grid and representation (vmr/number density and altitude/pressure) as the retrieval.

p.3 l.5: "Calculation of zonal averages over $L$ profiles . . . "
Why restrict to zonal?

p.3 l.12: "For a retrieval with $x_a = 0$ . . . "
This is a nitpick and I don't necessarily think it should be changed, but the same would be true with $x_a$=constant and a Tiknonov regularized retrieval. I guess the general condition would be if $x_a$ is in the null space of $R$.

p.3 l.22: "For a retrieval where an individual prior $x_a$ is used for each profile . . . "
I suppose this assumes that the prior used is a good representation of the true atmospheric state/variability.

p.3 l.15: "cov$(A, x)$ and be approximated by cov$(A, \hat{x})$"
I have a hard time intuitively understanding the implications of this approximation. I think that there are two things going on here, the first is the switch from the true state to the smoothed state, which I don't expect to have a large effect. But since the intention is to use this to compare two measurements, are we also assuming that both instruments have approximately equal sampling within whatever bin is being averaged?
p.4 l.8: "For retrievals performed in the log-space, all this becomes slightly more complicated ..."
It is fine to ignore the issues with log retrievals, since, as stated, averaging may have its own issues, but I have to wonder is this not a more general representation issue? Presumably if our goal was to compare a high resolution and a low resolution retrieval that both operated in log space, it would be possible using this framework if the averaging was done in log space.

p.4 l.10: eq. 12
Perhaps related to above, but this equation is hard to interpret when the $x$'s do not represent the same thing (some are in linear space some are logarithmic). Or maybe all the $x$'s are intended to be in linear space and the logarithm being applied to $x_{original}$ is missing?

p.7 l.12: "The covariance effects can exceed 10% and thus need to be considered when mean profiles are used for quantitative analysis and mean averaging kernels are applied."
This statement had me wondering about the implications of this effect beyond comparisons of two measurements. Say a data user is using zonally averaged MIPAS HCN data, but not actually applying any mean averaging kernel. Would having knowledge of the magnitude of this covariance term guide them in their analysis, similar to the way having a measure of vertical resolution from the averaging kernel would?

**Technical Comments**

p.4 l.18: eq. 13
Equation has extra equal signs.

p.7 l.3: "consistes"
consistes → consists

p.7 l.17: "we recommed"
recommed → recommend

---

## Author Comment (AC1) · 27 May 2019

The authors thank the reviewer for the helpful suggestions which will add clarity to the paper.

**Comment:** *Overview*
*The paper draws attention to a hitherto overlooked problem with the application of averaging kernel matrices, specifically that the AK matrix itself, A, has some dependence on the retrieved state, x. Hence, when using averaged data, <x> (e.g., a monthly mean), the appropriate averaging kernel <Ax> is not simply <A><x> constructed from an average AK matrix <A>.*

[Figure]

*Main Comments:*

*The paper is concise, well-argued and includes a suitable illustrative example, although I have some suggestions below as to how it might be further clarified so as to reach a wider audience...*

**Reply:** We thank the reviewer for the appreciation of our work.

**Comment:** *... While the main recommendation of the paper is that the data providers should also provide a correction term for the mean averaging kernel, it does seem more practical if, instead, the data providers themselves use the guidance in this paper to produce a suitable averaging kernel to accompany the averaged products. We all know that it is a struggle to get data user to understand and apply an averaging kernel, so we should avoid making their task any more complicated.*

**Reply:** The adequacy of the correction vector can be deduced with mathematical rigour. Since the correction vector is additive while a modified averaging kernel would modify the mean profile in a multiplicative manner, we doubt that it is possible to find a modified averaging kernel which does in effect the same as our suggested additive correction. We think that work with scientific data is best done in cooperation with the data providers, who will then help the data users to apply the averaging kernel and the correction term correctly.

**Comment:** *...I doubt if this paper will be the last word in the matter – there are a number of open issues which require a little more thought, such as logarithmic retrievals, retrievals of temperature/pointing/pressure, non-constant a priori data, averaging kernel matrices which are not square. However, this paper is a good starting*

*point for the conversation.*

**Reply:** We do agree that this paper can only be a first step.

**Comment:** *Minor Comments*

*1) The authors frequently resort to Latin. Personally I find it a welcome change from the usual stock phrases, although I expect some readers may not be quite so appreciative.*

**Reply:** We have gone through the manuscript and checked where it seemed appropriate to replace the latin terms by English ones.
*a priori* (various places): This is so common that it is not even printed in italic font in Copernicus journals; thus we have decided not to change it.
*viz.* (p1 l18): can be easily replaced by 'namely' without adding much length to the paper.
*mutatis mutandis* (p3 l11): although we found it also in an English dictionary, this term might indeed not be widely known. Can be replaced with 'in an analogue way'.
*cum grano salis* (p3 l17): We will replace this with "with some qualification..."
*a fortiori* (p8 l3): this is not so easy to replace so we consider to leave it unchanged.

**Comment:** *2) Abstract (and elsewhere): reference to 'covariance profile' although the suggested correction is a matrix rather than a profile.*

**Reply:** Here we respectfully disagree. The correction is a column vector. It results from a product of an $n \times n$ square matrix with a profile (see, e.g., Eq. 7).

**Comment:** *3) P1, L17: 'this is' does not make sense here.*

**Reply:** Will be replaced with 'that is to say'

**Comment:** *4) P1, L20: I don't think the use of monthly means requires any references, although no doubt Hegglin and Tegtmeier will appreciate being selected for multiple citations from among many, many such users.*

**Reply:** To our knowledge, the SPARC data initiative was the first large-scale international activity where the work was entirely monthly zonal means. And, beyond this, it was during the SPARC Data Initiave discussions where the title paper of the problem emerged. We agree that the original manuscript includes an over-exaggerated number of related references, and we will reduce the number of references to a single one (but we would, of course, not fight for leaving this reference included).

**Comment:** *5) P1, L20: suggest 'their' rather than 'her'.*

**Reply:** agreed.

**Comment:** *6) P2, L13: The casual reader may interpret this comment as suggesting none of this applies to non-linear, iterative retrievals, so I suggest rewording to emphasise that it still does.*

**Reply:** We agree. We will figure out some clearer wording here.

**Comment:** *7) P2, Eq 2: It could be pointed out that the main dependence on x in the AK matrix comes from the Jacobian matrix, K (although possibly also from R if some form of adaptive regularization is used), so whether or not there is any dependence of A on x is usually a consequence of whether or not K depends on x.*

**Reply:** Agreed. We will add such a statement after Eq (2) and modify the following statement because otherwise it might no longer be clear what 'this' refers to.

**Comment:** *8) P2, L25: An extra equation, $y - F(x\_a) = K(x - x\_a)$ would help the reader get from eq (2) to eq (3).*

**Reply:** Since $F(x)$ might be nonlinear, this extra equation does not necessarily hold in a general sense, although we see its didactic value. We consider to write something like "Within linear theory we have $y - F(x\_a) = K(x - x\_a)$ ..."

**Comment:** *9) P2, L28 onwards. This is confusing. Elsewhere averaging kernels are discussed as a characteristic of the lower-resolution (satellite) retrievals, but in this example (Eq 4) the averaging kernel seems to be on the grid of the higher resolution 'original' retrieval. Despite the similarity of Eq 4 and Eq 3, these seem to be two quite different things.*

**Reply:** We assume that the averaging kernel of the coarser resolved data set has been evaluated on a grid fine enough to do this transformation. For example, MIPAS averaging kernels are evaluated on a 1-km altitude grid although the MIPAS altitude resolution is typically only about 3 km. This allows application of Eq. 4 to, e.g., model data sampled on a 1-km grid. We realize that our terminology in the paper might lead the reader astray: Our $x_{original}$ is any high-resolved profile to be degraded. It is

NOT our original retrieval. We will change the terminology and use something like "high-resolution" to avoid this kind of misunderstanding. In the application described above, $x_{original}$ was meant to be the high-resolution model data to be degraded.

**Comment:** *10) P3, Eq 6 and elsewhere: if this is prepared with LaTeX, I suggest using nlangle and nrangle rather than < and > for the angle-brackets.*

**Reply:** ok, thanks!

**Comment:** *11) P3, Eq 7: it would be helpful to further simplify this here, giving $cov(A, x) = < Ax > - < A > < x >$ which makes Eq 6 clearer.*

**Reply:** Agreed, we will add this.

**Comment:** *12) It does not help that many of these equations are split over two lines, but that may not be the choice of the authors.*

**Reply:** This indeed looks odd in the one-column format of the discussion paper. But the final paper will be formatted in two columns, and to avoid errors in the equation when the manuscript will be transformed from a one-column format to a two-column format, where many of these equations have to be split over two lines, we have chosen a format for the equations which will be compatible also with the two-column style file.

**Comment:** *13) P3, L22: There seems to be more to be said than this simple phrase 'an individual prior $x\_a$'. For example, an individual but **almost** constant a priori could be used for each profile, in which case Eq (9) applies rather than Eq (11).*

*The key is obviously what sort of 'individual prior' leads to the two covariances being approximately equal.*

**Reply:** We had two different applications in mind:

1. The retrieval scientist might use climatologies as a priori. This is often done, and Rodgers' account of the smoothing error seems to suggest that this kind of priori information can be used. In this case Eq(9) applies.

2. In some other cases, the best available prior information for each individual retrieval can be used. For example, for MIPAS we use meteorological analyses as a priori for temperature retrievals (we constrain only the shape, not the values, but that is another story...). Eq (10) has been tailored for this kind of applications.

We agree that the entire 'grey scale' between these extremes deserves some discussion. We will expand on this in the text.

**Comment:** *14) P4, Eq (11): In this case I think the extra equation confuses (especially when split over multiple lines) rather than clarifies. Perhaps better to refer back to Eq (6) and simply state the simplified result.*

**Reply:** The purpose of Eq (11) is to make clear which of the terms cancel out with the given assumptions in force. This is not easily seen in Eq (6).

**Comment:** *15) P4, L26: nbar r seems to be introduced in the wrong place in this sentence, presumably it should be after 'normalized covariance term'*

[Figure]

**Reply:** yes, indeed, thanks for spotting.

**Comment:** *16) P4, L14: I'm surprised that this produces stable results, eg for HCN at higher altitudes, where the <x> in the denominator would tend to zero. Covariance terms, as in the Pearson correlation, are usually scaled by the square root of the variance, so don't have this problem.*

**Reply:** well, rather by the product of both standard deviations involved rather than the variance, but we see your point. Our normalized covariance can be interpreted as a relative error. It is admittedly unstable, similarly as relative or percentage errors in the case of small measured values (and in this case nobody complains). A normalized quantity calculated along the idea of Pearson's correlation coefficient would be more stable but it would require an entirely different interpretation. We will test your suggestion and, if successful, will use it either instead or in addition to our normalized covariance.

**Comment:** *17) P5, L17: 'recommend' (spelling).-*

**Reply:** Thanks for spotting.

---

## Author Comment (AC2) · 27 May 2019

The authors thank both reviewer for the helpful suggestions which will add clarity to the paper.

**Comment:** *This manuscript discusses an important and often ignored issue involving the application of averaging kernels to mean profiles. A solution to the problem is presented where the covariance between the averaging kernel and the atmospheric state is calculated. Examples are shown applying the method to MIPAS, and recommendations are given to data producers of monthly zonal mean data.*
*The manuscript is well written and suitable for publication in AMT after a few comments are taken into account.*

[Figure]

**Reply:** We thank the reviewer for this positive evaluation.

**Comment:** *General Comments*
*The discussion and conclusion (including the recommendations) of the paper focuses on the ideal case where the data producer actually calculates (and stores) an averaging kernel for each individual profile. It is somewhat common to only produce representative averaging kernels and perhaps use them as a metric for retrieval performance in a validation/retrieval paper or data quality document. Would a possible recommendation of this work be that a few of these covariance terms should be calculated and included as an assessment of the data quality?*

**Reply:** These covariance terms refer to mean averaging kernels. We have no idea to which degree they are applicable to representative individual averaging kernels. Since the covariance terms depend on the ensemble over which it is averaged, and since it may not be known in advance what kind of averages the data user wants, it will not be easily possible to produce useful covariance profiles in advance. Instead one might consider to calculate the averaging kernels for each retrieval (which is not so much additional effort) and to calculate zonal mean profiles and averaging kernels immediately as the last step of each individual retrieval. In this case one would have the mean averaging kernels and the covariances without storing each averaging kernel. They can be deleted immediately after their consideration for the mean values.

**Comment:** *Related to the above point, I have to wonder, is the covariance profile useful beyond a correction when applying the mean averaging kernel? My (perhaps wrong) interpretation is that when the covariance profile is 0, the mean of the retrieved profile is a smoothed version of the true mean atmospheric state. I suppose what I*

[Figure]

*am asking is that if the covariance profile is not 0, is it wrong to interpret the retrieved mean as a smoothed version of the true atmospheric mean? If so, I would like to see a discussion of this included in the manuscript.*

**Reply:** We are not aware of any other interpretation beyond the one offered. The interpretation that zero covariance means that the mean of the retrieved profiles is a smoothed version of the true mean atmospheric state is not true, at least not in a general sense. Assume a case with infinite noise, i.e., no measurement information. The retrieval will then be identical to the a priori. Assume further that a constant (e.g. climatological) a priori has been chosen. The the result will not vary at all. Thus also the covariance will be zero although the mean result is fully determine by the a priori, in shape and values.

**Comment:** *Minor Comments*
*p.1 l.9: ". . . on a given altitude grid . . . "*
*Here and throughout this section it is written that altitude is the vertical coordinate, however all of the arguments should equally apply to any vertical coordinate.*

**Reply:** With "altitude" we mean any vertical coordinate, not only geometrical altitude. We will change the wording to make this clear.

**Comment:** *p.2 l.18: "For a constrained retrieval of the type" The way this is presented the reader may assume that what follows only applies to retrievals applying a (possibly iterative) form of eq. 1, when the concepts here are more general.*

**Reply:** We agree, but this type or retrieval is the only one for which averaging kernels are reported at all. But you are right, in principle our arguments hold for

any other type of retrievals if averaging kernels are made available. These can be evaluated, e.g., by perturbation studies. We will try to reword the text to make this clear.

**Comment:** *p.2 l.29: eq. 4: Somewhere here I would like to see a brief mention that $x_o riginal$ needs to be converted to the same grid and representation (vmr/number density and altitude/pressure) as the retrieval.*

**Reply:** agreed.

**Comment:** *p.3 l.5: "Calculation of zonal averages over L profiles . . . " Why restrict to zonal?*

**Reply:** This is meant as an example. We will add 'e.g.'.

**Comment:** *p.3 l.12: "For a retrieval with $x_a$ = 0 . . . " This is a nitpick and I don't necessarily think it should be changed, but the same would be true with $_xa$=constant and a Tiknonov regularized retrieval. I guess the general condition would be if $x_a$ is in the null space of R.*

**Reply:** ok, indeed for an altitude-constant prior and an A with unity measurement response in all altitudes, we have $< A >< x_a >=< x_a >$, which cancels with the first $< x_a >$ term. And a covariance involving a time-constant $x_a$ will also be zero. We will think about a wording which makes a more general statement without being too complicated.

**Comment:** *p.3 l.22: "For a retrieval where an individual prior xa is used for each profile*

[Figure]

*. . . " I suppose this assumes that the prior used is a good representation of the true atmospheric state/variability.*

**Reply:** Yes, this is indeed a precondition for Eq 10. We will mention this.

**Comment:** *p.3 l.15: "cov(A; x) and be approximated by $cov(A; \hat{x})$" I have a hard time intuitively understanding the implications of this approximation. I think that there are two things going on here, the first is the switch from the true state to the smoothed state, which I don't expect to have a large effect. But since the intention is to use this to compare two measurements, are we also assuming that both instruments have approximately equal sampling within whatever bin is being averaged?*

**Reply:** Although there exists literature on comparison between instruments where the averaging kernels of both instruments are considered (Rodgers and Connor, 2003), the approaches most often taken are to do a direct comparison (without using averaging kernels) as long as the vertical resolutions are similar and the measurement response is large, and to apply the averaging kernel of the coarser resolved retrieval to the result of the better resolved retrieval, if the contrast in resolution justifies this (Connor et al. 1995). In none of these cases we have to deal with averaging kernels of both instruments, thus no such assumption is necessary. The same holds for model-measurement comparisons. Here the averaging kernel of the measurement is typically applied to the model output. Also in this case no such assumption is necessary.

**Comment:** *p.4 l.8: "For retrievals performed in the log-space, all this becomes slightly more complicated . . . " It is fine to ignore the issues with log retrievals, since, as stated, averaging may have its own issues, but I have to wonder is this not a more*

*general representation issue? Presumably if our goal was to compare a high resolution and a low resolution retrieval that both operated in log space, it would be possible using this framework if the averaging was done in log space.*

**Reply:** Averaging in the log-space typically does not solve related problems; particularly it will not remove biases introduced by the retrieval in the log space (see, Funke and von Clarmann, 2012).

**Comment:** *p.4 l.10: eq. 12 Perhaps related to above, but this equation is hard to interpret when the x's do not represent the same thing (some are in linear space some are logarithmic). Or maybe all the x's are intended to be in linear space and the logarithm being applied to xoriginal is missing?*

**Reply:** The latter is the case; it should read $Ax_{original}$. Thanks for spotting!

**Comment:** *p.7 l.12: "The covariance effects can exceed 10% and thus need to be considered when mean profiles are used for quantitative analysis and mean averaging kernels are applied." This statement had me wondering about the implications of this effect beyond comparisons of two measurements. Say a data user is using zonally averaged MIPAS HCN data, but not actually applying any mean averaging kernel. Would having knowledge of the magnitude of this covariance term guide them in their analysis, similar to the way having a measure of vertical resolution from the averaging kernel would?*

**Reply:** We do not have any idea how to use the covariance term for other purposes than that described in the paper.

**Comment:** *Technical Comments*
*p.4 l.18: eq. 13 Equation has extra equal signs.*

**Reply:** Thanks for spotting; they will be removed.

**Comment:** *p.7 l.3: "consistes" consistes ! consists*

**Reply:** Typo will be corrected

**Comment:** *p.7 l.17: "we recommed" recommed ! recommend*

***Reply:** Typo will be corrected. Thanks for spotting.*
* * *
*Interactive comment on Atmos. Meas. Tech. Discuss., doi:10.5194/amt-2019-61, 2019.*

---

## Author Response (AR1)

The authors thank both reviewers for their careful reading of the manuscript and the helpful suggestions which have added clarity to the paper.

**Review 1**

**Comment:** *Overview*
*The paper draws attention to a hitherto overlooked problem with the application of averaging kernel matrices, specifically that the AK matrix itself, A, has some dependence on the retrieved state, x. Hence, when using averaged data, ¡x¿ (e.g., a monthly mean), the appropriate averaging kernel ¡Ax¿ is not simply ¡A¿¡x¿ constructed from an average AK matrix ¡A¿.*

*Main Comments:*

*The paper is concise, well-argued and includes a suitable illustrative example, although I have some suggestions below as to how it might be further clarified so as to reach a wider audience...*

**Reply:** We thank the reviewer for the appreciation of our work.

**Action:** See actions in reply to the specific comments.

**Comment:** *... While the main recommendation of the paper is that the data providers should also provide a correction term for the mean averaging kernel, it does seem more practical if, instead, the data providers themselves use the guidance in this paper to produce a suitable averaging kernel to accompany the averaged products. We all know that it is a struggle to get data user to understand and apply an averaging kernel, so we should avoid making their task any more complicated.*

**Reply:** The adequacy of the correction vector can be deduced with mathematical rigour. Since the correction vector is additive while a modified averaging kernel would modify the mean profile in a multiplicative manner, we doubt that it is possible to find a modified averaging kernel which does in effect the same as our suggested additive correction. We think that work with scientific data is best done in cooperation with the data providers, who will then help the data users to apply the averaging kernel and the correction term correctly.

**Action:** None

**Comment:** *...I doubt if this paper will be the last word in the matter – there are a number of open issues which require a little more thought, such as logarithmic retrievals, retrievals of temperature/pointing/pressure, non-constant a priori data, averaging kernel matrices which are not square. However, this paper is a good starting point for the conversation.*

**Reply:** We do agree that this paper can only be a first step.

**Action:** None

**Comment:** *Minor Comments*

*1) The authors frequently resort to Latin. Personally I find it a welcome change from the usual stock phrases, although I expect some readers may not be quite so appreciative.*

**Reply:** We have gone through the manuscript and checked where it seemed appropriate to replace the latin terms by English ones.

**Action:**
*a priori* (various places): This is so common that it is not even printed in italic font in Copernicus journals; thus we have decided not to change it.
*viz.* (p1 l18): has been replaced with 'namely'.
*mutatis mutandis* (p3 l11): has been replaced with 'in an analogue way'.
*cum grano salis* (p3 l17): has been replaced with "with some qualification..."
*a fortiori* (p8 l3): has been replaved with 'even more'.

**Comment:** *2) Abstract (and elsewhere): reference to 'covariance profile' although the suggested correction is a matrix rather than a profile.*

**Reply:** Here we respectfully disagree. The correction is a column vector. It results from a product of an $n \times n$ square matrix with a profile (see, e.g., Eq. 7).

**Action:** None

**Comment:** *3) P1, L17: 'this is does not make sense here.*

**Reply:** Agreed.

**Action:** Has been replaced with 'that is to say'

**Comment:** *4) P1, L20: I don't think the use of monthly means requires any references, although no doubt Hegglin and Tegtmeier will appreciate being selected for multiple citations from among many, many such users.*

**Reply:** To our knowledge, the SPARC data initiative was the first large-scale international activity where the work was entirely monthly zonal means. And, beyond this, it was during the SPARC Data Initaive discussions where the title paper of the problem emerged. We agree that the original manuscript includes an over-exaggerated number of related references.

**Action:** The number of references has been reduced to a single one.

**Comment:** *5) P1, L20: suggest 'their rather than her.*

**Reply:** agreed.

**Action:** Changed as suggested.

**Comment:** *6) P2, L13: The casual reader may interpret this comment as suggesting none of this applies to non-linear, iterative retrievals, so I suggest rewording to emphasise that it still does.*

**Reply:** We agree.

**Action:** A clarifying sentence has been added.

**Comment:** *7) P2, Eq 2: It could be pointed out that the main dependence on $\vec{x}$ in the AK matrix comes from the Jacobian matrix, $\mathbf{K}$ (although possibly also from $\mathbf{R}$ if some form of adaptive regularization is used), so whether or not there is any dependence of $\mathbf{A}$ on $\vec{x}$ is usually a consequence of whether or not $\mathbf{K}$ depends on $\vec{x}$.*

**Reply:** Agreed.

**Action:** A statement has been added after Eq (2) and the following statement has been slightly modified because otherwise it might no longer be clear what 'this' refers to.

**Comment:** *8) P2, L25: An extra equation, $\vec{y} - \mathbf{F}(\vec{x}_a) = \mathbf{K}(\vec{x} - \vec{x}_a)$ would help the reader get from eq (2) to eq (3).*

**Reply:** Since $\mathbf{F}(\vec{x})$ might be nonlinear, this extra equation does not necessarily hold in a general sense, although we see its didactic value.

**Action:** We have added: With the averaging kernel matrix introduced above, and using the linearization

$$\vec{y} - \mathbf{F}(\vec{x}_{\mathrm{a}}) \approx \mathbf{K}(\vec{x} - \vec{x}_{\mathrm{a}}), \tag{1}$$

**Comment:** *9) P2, L28 onwards. This is confusing. Elsewhere averaging kernels are discussed as a characteristic of the lower-resolution (satellite) retrievals, but in this example (Eq 4) the averaging kernel seems to be on the grid of the higher resolution 'original' retrieval. Despite the similarity of Eq 4 and Eq 3, these seem to be two quite different things.*

**Reply:** We assume that the averaging kernel of the coarser resolved data set has been evaluated on a grid fine enough to do this transformation. For example, MIPAS averaging kernels are evaluated on a 1-km altitude grid although the MIPAS altitude resolution is typically only about 3 km. This allows application of Eq. 4 to, e.g., model data sampled on a 1-km grid. We realize that our terminology in the paper might lead the reader astray: Our $x_{original}$ is any high-resolved profile to be degraded. It is NOT our original retrieval. In the application described above, $x_{original}$ was meant to be the high-resolution model data to be degraded.

**Action:** We have changed our terminology and use the subscript "high-resolution" to avoid this kind of misunderstanding. We have also added some prose for clarification.

**Comment:** *10) P3, Eq 6 and elsewhere: if this is prepared with LaTeX, I suggest using nlangle and nrangle rather than $<$ and $>$ for the angle-brackets.*

**Reply:** ok, thanks!

**Action:** done as suggested.

**Comment:** *11) P3, Eq 7: it would be helpful to further simplify this here, giving* $\mathrm{cov}(\mathbf{A}, \tilde{x}) = \langle \mathbf{A}\tilde{x} \rangle - \langle \mathbf{A} \rangle \langle \tilde{x} \rangle$ *which makes Eq 6 clearer.*

**Reply:** Agreed.

**Action:** Done as suggested.

**Comment:** *12) It does not help that many of these equations are split over two lines, but that may not be the choice of the authors.*

**Reply:** This indeed looks odd in the one-column format of the discussion paper. But the final paper will be formatted in two columns, and to avoid errors in the equations when the manuscript will be transformed from a one-colum format to a two-column format, where many of these equations have to be split over two lines, we have chosen a format for the equations which will be compatible also with the two-column style file.

**Action:** None.

**Comment:** *13) P3, L22: There seems to be more to be said than this simple phrase 'an individual prior x_a'. For example, an individual but* **almost** *constant a priori could be used for each profile, in which case Eq (9) applies rather than Eq (11). The key is obviously what sort of 'individual prior' leads to the two covariances being approximately equal.*

**Reply:** We had two different applications in mind:

1. The retrieval scientist might use climatologies as a priori. This is often done, and Rodgers' account of the smoothing error seems to suggest that this kind of priori information can be used. In this case Eq(9) applies.

2. In some other cases, the best available prior information for each individual retrieval can be used. For example, for MIPAS we use meteorological analyses as a priori for temperature retrievals (we constrain only the shape, not the values, but that is another story...). Eq (10) has been tailored for this kind of applications.

We agree that the entire 'grey scale' between these extremes deserves some discussion.

**Action:** Some text has been added.

**Comment:** *14) P4, Eq (11): In this case I think the extra equation confuses (especially when split over multiple lines) rather than clarifies. Perhaps better to refer back to Eq (6) and simply state the simplified result.*

**Reply:** The purpose of Eq (11) is to make clear which of the terms cancel out with the given assumptions in force. This is not easily seen in Eq (6).

**Action:** None.

**Comment:** *15) P4, L26: nbar r seems to be introduced in the wrong place in this sentence, presumably it should be after 'normalized covariance term'*

**Reply:** yes, indeed, thanks for spotting.

**Action:** corrected.

**Comment:** *16) P4, L14: I'm surprised that this produces stable results, eg for HCN at higher altitudes, where the ¡x¿ in the denominator would tend to zero. Covariance terms, as in the Pearson correlation, are usually scaled by the square root of the variance, so don't have this problem.*

**Reply:** Well, rather by the product of both standard deviations involved than by the variance, but we see your point. Our normalized covariance can be interpreted as a relative error. It is admittedly unstable, similarly as relative or percentage errors in the case of small measured values (and in the latter case nobody complains). A normalized quantity calculated along the idea of Pearson's correlation coefficient would be more stable but it would require an entirely different interpretation. Since we are interested in the relative error related with

ignoring the covariance effect, we think that our variant is more adequate.

**Action:** None.

**Comment:** *17) P5, L17: 'recommend' (spelling).-*

**Reply:** Thanks for spotting.

**Action:** Corrected.

**Review 2**

**Comment:** *This manuscript discusses an important and often ignored issue involving the application of averaging kernels to mean profiles. A solution to the problem is presented where the covariance between the averaging kernel and the atmospheric state is calculated. Examples are shown applying the method to MIPAS, and recommendations are given to data producers of monthly zonal mean data.*
*The manuscript is well written and suitable for publication in AMT after a few comments are taken into account.*

**Reply:** We thank the reviewer for this positive evaluation.

**Action:** See actions in reply to the specific comments.

**Comment:** *General Comments*
*The discussion and conclusion (including the recommendations) of the paper focuses on the ideal case where the data producer actually calculates (and stores) an averaging kernel for each individual profile. It is somewhat common to only produce representative averaging kernels and perhaps use them as a metric for retrieval performance in a validation/retrieval paper or data quality document. Would a possible recommendation of this work be that a few of these covariance terms should be calculated and included as an assessment of the data quality?*

**Reply:** These covariance terms refer to mean averaging kernels. We have no idea to which degree they are applicable to representative individual averaging kernels. Since the covariance terms depend on the ensemble over which it is averaged, and since it may not be known in advance what kind of averages the data user wants, it will not be easily possible to produce useful covariance profiles in advance. Instead one might consider to calculate the averaging kernels for each retrieval (which is not so much additional effort) and to calculate zonal mean profiles and averaging kernels immediately as the last step of each individual retrieval. In this case one would have the mean averaging kernels and the covariances without storing each averaging kernel. They can be deleted immediately after their consideration for the mean values.

**Action: None**

**Comment:** *Related to the above point, I have to wonder, is the covariance profile useful beyond a correction when applying the mean averaging kernel? My (perhaps wrong) interpretation is that when the covariance profile is 0, the mean of the retrieved profile is a smoothed version of the true mean atmospheric state. I suppose what I am asking is that if the covariance profile is not 0, is it wrong to interpret the retrieved mean as a smoothed version of the true atmospheric mean? If so, I would like to see a discussion of this included in the manuscript.*

**Reply:** We are not aware of any other interpretation beyond the one offered. The interpretation that zero covariance means that the mean of the retrieved profiles is a smoothed version of the true mean atmospheric state is not true, at least not in a general sense. Assume a case with infinite noise, i.e., no measurement information. The retrieval will then be identical to the a priori. Assume further that a constant (e.g. climatological) a priori has been chosen. The result will not vary at all. Thus also the covariance will be zero although the mean result is fully determined by the a priori, in shape and values.

**Action: None**

**Comment:** *Minor Comments*
*p.1 l.9: ". . . on a given altitude grid . . . "*
*Here and throughout this section it is written that altitude is the vertical coordinate, however all of the arguments should equally apply to any vertical coordinate.*

**Reply:** With "altitude" we mean any vertical coordinate, not only geometrical altitude.

**Action:** We have added a footnote explaining that we mean altitude in a more general sense, not limited to geometrical altitude.

**Comment:** *p.2 l.18: "For a constrained retrieval of the type" The way this is presented the reader may assume that what follows only applies to retrievals applying a (possibly iterative) form of eq. 1, when the concepts here are more general.*

**Reply:** We agree, but this type or retrieval is the only one for which averaging kernels are reported at all. But you are right, in principle our arguments hold for any other type of retrievals if averaging kernels are made available. These can be evaluated, e.g., by perturbation studies.

**Action:** We have added "...or any equivalent formulation of it."
**Comment:** *p.2 l.29: eq. 4: Somewhere here I would like to see a brief mention that $x_{o}riginal$ needs to be converted to the same grid and representation*

*(vmr/number density and altitude/pressure) as the retrieval.*

**Reply:** agreed.

**Action:** we have added "It goes without saying that the high-resolved profile has to be resampled on the grid on which the application of the averaging kernel is performed, and, if applicable, transformed to the same units (volume mixing ratio, number density, etc."

**Comment:** *p.3 l.5: "Calculation of zonal averages over L profiles . . . " Why restrict to zonal?*

**Reply:** This is meant as an example.

**Action:** We have added 'e.g.'.

**Comment:** *p.3 l.12: "For a retrieval with $\vec{x}_a = 0$ . . . " This is a nitpick and I dont necessarily think it should be changed, but the same would be true with $\vec{x}_a$=constant and a Tiknonov regularized retrieval. I guess the general condition would be if $\vec{x}_a$ is in the null space of $\mathbf{R}$.*

**Reply:** Ok, indeed for an altitude-constant prior and an $\mathbf{A}$ with unity measurement response in all altitudes, we have $\langle\mathbf{A}\rangle\langle\vec{x}_{\mathrm{a}}\rangle = \langle\vec{x}_{\mathrm{a}}\rangle$, which cancels with the first $\langle\vec{x}_{\mathrm{a}}\rangle$ term. And a covariance involving a time-constant $\vec{x}_{\mathrm{a}}$ will also be zero.

**Action:** We have added "The same is true if for all retrievals the same altitude-constant prior is used in combination with an averaging kernel with unity row sums as associated with purely smoothing constraints.".

**Comment:** *p.3 l.22: "For a retrieval where an individual prior $\vec{x}|texta$ is used for each profile . . . " I suppose this assumes that the prior used is a good representation of the true atmospheric state/variability.*

**Reply:** Yes, this is indeed a precondition for Eq 10.

**Action:** We have added "...i.e., that the prior information is a good representation of the true atmospheric state and variability."

**Comment:** *p.3 l.15: "cov$(\mathbf{A}; \vec{x})$ and be approximated by cov$(\mathbf{A}; \hat{\vec{x}})$" I have a hard time intuitively understanding the implications of this approximation. I think that there are two things going on here, the first is the switch from the true state to the smoothed state, which I don't expect to have a large effect. But since the intention is to use this to compare two measurements, are we also assuming that both instruments have approximately equal sampling within whatever bin is being averaged?*

**Reply:** Yes, we assume either identical or representative sampling. If the sampling is representative with respect to mean and variance, then different sampling should not be an issue.

**Action:** None

**Comment:** *p.4 l.8: "For retrievals performed in the log-space, all this becomes slightly more complicated . . . " It is fine to ignore the issues with log retrievals, since, as stated, averaging may have its own issues, but I have to wonder is this not a more general representation issue? Presumably if our goal was to compare a high resolution and a low resolution retrieval that both operated in log space, it would be possible using this framework if the averaging was done in log space.*

**Reply:** Averaging in the log-space typically does not solve related problems; particularly it will not remove biases introduced by the retrieval in the log space (see, Funke and von Clarmann, 2012).

**Action:** None.

**Comment:** *p.4 l.10: eq. 12 Perhaps related to above, but this equation is hard to interpret when the x's do not represent the same thing (some are in linear space some are logarithmic). Or maybe all the x's are intended to be in linear space and the logarithm being applied to xoriginal is missing?*

**Reply:** The latter is the case; it should read $Ax_{original}$. Thanks for spotting!

**Action:** Corrected.

**Comment:** *p.7 l.12: "The covariance effects can exceed 10% and thus need to be considered when mean profiles are used for quantitative analysis and mean averaging kernels are applied." This statement had me wondering about the implications of this effect beyond comparisons of two measurements. Say a data user is using zonally averaged MIPAS HCN data, but not actually applying any mean averaging kernel. Would having knowledge of the magnitude of this covariance term guide them in their analysis, similar to the way having a measure of vertical resolution from the averaging kernel would?*

**Reply:** We do not have any idea how to use the covariance term for other purposes than that described in the paper.

**Action:** None.

**Comment:** *Technical Comments*
*p.4 l.18: eq. 13 Equation has extra equal signs.*

**Reply:** Thanks for spotting

**Action:** Corrected.

**Comment:** *p.7 l.3: "consistes" consistes ! consists*

**Reply:** Thanks for spotting!

**Action:** corrected.

**Comment:** *p.7 l.17: "we recommed" recommed ! recommend*
**Reply:** *Thanks for spotting.*

**Action:** *Corrected.*

[revised manuscript text omitted]

&= \underline{<}\langle \boldsymbol{x}_{\mathrm{a}} \underline{>}\rangle - \underline{<}\langle \mathbf{A} \underline{>}\rangle\underline{<}\langle \boldsymbol{x}_{a} \underline{>}\rangle - \\
&\quad \mathrm{cov}(\mathbf{A}, \boldsymbol{x}_{\mathrm{a}}) + \underline{<}\langle \mathbf{A} \underline{>}\rangle\underline{<}\langle \boldsymbol{x} \underline{>}\rangle + \mathrm{cov}(\mathbf{A}, \boldsymbol{x}),
\end{aligned}
\tag{8}
$$

where

$$\mathrm{cov}(\mathbf{A}, \boldsymbol{x}) = \frac{1}{L}\sum_{l=1}^{L}(\mathbf{A}_l - \underline{<}\langle \mathbf{A} \underline{>}\rangle)(\boldsymbol{x}_l - \underline{<}\langle \boldsymbol{x} \underline{>}\rangle), \tag{9}$$

and in an analogue way for $\mathrm{cov}(\mathbf{A}, \boldsymbol{x}_{\mathrm{a}})$.
* * *
[2]Here a caveat is in order. The average of profiles which are 'optimal' in the sense of maximum a posteriori information and where the a priori information is the same for all averaged profiles is not the optimal average. This is, because the weight of the a priori information will be too large in the average. A more thorough discussion of this issue, however, is beyond the scope of this paper.

For a retrieval with $\boldsymbol{x}_\mathrm{a} = \boldsymbol{0}$ (or $\boldsymbol{x}_\mathrm{a}$ constant with altitude and a purely smoothing constraint) this simplifies to

[revised manuscript text omitted]

---

## Referee Report (RR1)

The authors have generally addressed my original comments sufficiently, there is just one slight point I would like to comment on. My original comment and the author's reply:

**Comment**: Related to the above point, I have to wonder, is the covariance profile useful beyond a correction when applying the mean averaging kernel? My (perhaps wrong) interpretation is that when the covariance profile is 0, the mean of the retrieved profile is a smoothed version of the true mean atmospheric state. I suppose what I am asking is that if the covariance profile is not 0, is it wrong to interpret the retrieved mean as a smoothed version of the true atmospheric mean? If so, I would like to see a discussion of this included in the manuscript.

**Reply**: We are not aware of any other interpretation beyond the one offered. The interpretation that zero covariance means that the mean of the retrieved profiles is a smoothed version of the true mean atmospheric state is not true, at least not in a general sense. Assume a case with infinite noise, i.e., no measure ment information. The retrieval will then be identical to the a priori. Assume further that a constant (e.g. climatological) a priori has been chosen. The result will not vary at all. Thus also the covariance will be zero although the mean result is fully determined by the a priori, in shape and values.

The author's are certainly correct with their example, but I think the confusion comes from my (mis)use of the word "smoothing". If you take Eq. 6 from the manuscript and set the covariance terms to 0, you obtain

$$\langle \hat{x} \rangle = \langle x_a \rangle + \langle A \rangle (\langle x \rangle - \langle x_a \rangle).$$

This is the same as the standard "smoothing" equation that you would get for a single profile, with all of the quantities replaced by their means. So the point that I was trying to make is that if the covariance terms are zero there is an analogous interpretation for the mean profile compared to a single profile, which is no longer the case if the covariance terms are non-zero. I think that a statement to this effect in the manuscript would be helpful for the reader to better understand the implications of the covariance terms being zero or non-zero.

---

## Author Response (AR2)

The authors thank both reviewers for their favorable reception of the revised manuscript. There was only one open issue in the recevt version:

**Original Comment:** Related to the above point, I have to wonder, is the covariance profile useful beyond a correction when applying the mean averaging kernel? My (perhaps wrong) interpretation is that when the covariance profile is 0, the mean of the retrieved profile is a smoothed version of the true mean atmospheric state. I suppose what I am asking is that if the covariance profile is not 0, is it wrong to interpret the retrieved mean as a smoothed version of the true atmospheric mean? If so, I would like to see a discussion of this included in the manuscript.

**Original Reply:** We are not aware of any other interpretation beyond the one offered. The interpretation that zero covariance means that the mean of the retrieved profiles is a smoothed version of the true mean atmospheric state is not true, at least not in a general sense. Assume a case with infinite noise, i.e., no measurement information. The retrieval will then be identical to the a priori. Assume further that a constant (e.g. climatological) a priori has been chosen. The result will not vary at all. Thus also the covariance will be zero although the mean result is fully determined by the a priori, in shape and values.

**New Comment:** The author's are certainly correct with their example, but I think the confusion comes from my (mis)use of the word "smoothing". If you take Eq. 6 from the manuscript and set the covariance terms to 0, you obtain

$$\langle \hat{\vec{x}} \rangle = \langle \vec{x}_a \rangle + \langle \mathbf{A} \rangle (\langle \vec{x} \rangle - (\langle \vec{x}_a \rangle)).$$

This is the same as the standard "smoothing" equation that you would get for a single profile, with all of the quantities replaced by their means. So the point that I was trying to make is that if the covariance terms are zero there is an analogous interpretation for the mean profile compared to a single profile, which is no longer the case if the covariance terms are non-zero. I think that a statement to this effect in the manuscript would be helpful for the reader to better understand the implications of the covariance terms being zero or non-zero.

**New Reply:** Ok, now we've got it. The reviewer is obviously right. When writing the paper this issue was so obvious to us that we simply forgot to mention is explicitly but we agree that a statement on this should be made.

**Action:** We have included the following paragraph after Eq. (9):

[revised manuscript text omitted]